# Application of Performance and Efficiency Indicators in Measuring the Level of Success of Public Universities in Poland

Ewa Multan [1] , Marzena Wójcik-Augustyniak [1], Bartosz Sobotka [2] and Jakub Bis [3],*

[1] Faculty of Social Sciences, University of Natural Sciences and Humanities, 39 Zytnia Street, 08-110 Siedlce, Poland; ewa.multan@uph.edu.pl (E.M.); marzena.wojcik-augustyniak@uph.edu.pl (M.W.-A.)
[2] University College of Enterprise and Administration in Lublin, Bursaki 12, 20-150 Lublin, Poland; b.sobotka@sektorowaradanub.pl
[3] Faculty of Management, Lublin University of Technology, Nadbystrzycka 38d, 20-618 Lublin, Poland
* Correspondence: j.bis@pollub.pl

**Abstract:** Success can be interpreted and assessed in various ways. This article proposes evaluating university success through performance and efficiency indicators, inspired by the Positioning School of Strategy and New Public Management. This approach faces challenges, such as limited economic data for Polish public universities. The article aimed to identify factors and success levels for public universities in the higher education sector. The research question, "What are the measures and levels of success of public universities?", was divided into three specific inquiries: key success factors, performance and efficiency indicators, and success levels for selected Polish public universities. The study involved analysing international and Polish university rankings, interviewing 53 public university experts in Poland, and examining efficiency indicators for 10 selected public universities. Critical success factors (research excellence, education excellence, international cooperation) and critical performance indicators were identified. Efficiency indicators demonstrated the good financial condition of selected institutions. Critical success factors and critical performance indicators were defined, and performance and efficiency measures were used to assess the success of Polish public higher education institutions. The authors acknowledge the need for parametric and non-parametric methods to fully evaluate success. The proposed performance-oriented measurement tool can assist public university leaders in making strategic decisions.

**Keywords:** success; critical success factors; critical performance indicators; efficiency; universities in Poland

## 1. Introduction

It is difficult to give a clear answer to the questions, "What is success, and how do we measure it?", owing to the fact that success may be considered in multiple dimensions and by various approaches (personal, professional, financial, image (PR), sales, organisational, commercial success, etc.). For each organisation, success may mean a completely different thing—for one, success may be entering a new market, starting a new type of activity, surviving in the market, or maintaining a top market position. Moreover, what is seen as a success by one organisation may be perceived as a failure by another. That is why it is so difficult to specify precisely what success is and how it may be measured.

As a consequence of the assumptions of the Positioning School of Strategy that profit may be considered the quantitative measure of a company's success [1], the expectations arising from the New Public Management in which "higher education institutions are expected to account for their performance" [2], and because of the conditions in which modern universities operate, forcing continuous monitoring and evaluation of the performance of both the entire university organisational system and the evaluation of the performance of structural units and employees [3], for the purposes of this study, profit

and performance as clear/hard/measurable criteria were adopted as measures of success for a public university.

Unfortunately, in the case of the higher education sector, the main obstacle in measuring the effectiveness of universities is the unavailability of economic data of public universities in Poland. A clearer microdata policy at the national level is, therefore, needed.

In this article, the authors conducted an analysis of the indicators of university functioning performance, published in The Perspektywy University Ranking [4], and the analysis of the efficiency levels of selected public universities in Poland in the scope of university finance management (based on balance sheets as of 2019 for the 10 selected universities) with the use of return on assets (ROA) and return on equity (ROE), which were juxtaposed in the prepared matrix of the success level.

In the management literature and practice, we use so-called key performance indicators (KPIs), which according to J. Reh are the "values used to monitor and measure performance" that most industries have [5]. KPIs are the tools that are needed to understand and measure success [6]; they are "financial and non-financial indicators used as yardsticks in the process of measuring organisational performance" [7].

The main purpose of the article was to highlight the factors by which the success of public universities can be measured and the levels of success that can be achieved in the HEI sector. The main research problem, "What are the measures and levels of success of public universities?", was broken into three research questions:

RQ1: Which key success factors can be considered critical to the success of public universities?

RQ2: Can performance (CPI) and efficiency (ROA, ROE) indicators be considered as measures of the success of public universities?

RQ3: What are the levels of success of the selected public universities in Poland?

Although the authors in this article focused on parametric indicators, they are aware that non-parametric (e.g., DEA) and parametric (e.g., SFA) methods should be used to fully evaluate the success of a university.

This article is structured as follows: Section 1 describes the background of the literature on the FSC and CSF practices and the ROA and ROE financial ratios. Section 2 presents the next stages of the research procedure. Section 3 describes the results obtained, such as the CSF and CPI identification model, as well as the results and effectiveness of the analysed universities. Section 4 emphasizes the comparative discussion between the results obtained and the international literature and illustrates the main implications. The article ends with the authors' proposal of a matrix of the success level.

## 2. Literature Review

Due to multidimensional approaches, the literature offers numerous definitions of success. However, it may be assumed that, most frequently, success is referred to as achieving a set objective/objectives [8] or desired goals [9] and is "both particular, against specific objectives, and subjective—in the sense of who selects which goals and which performance benchmarks" [10]. Moreover, the way personal failure is defined and an individual's competitive attitude will also reflect the concept of success [9]. The manner in which success is defined depends on key issues considered as success factors, as well as the parameters and their levels adopted as a condition for success [11]. It may also be stated that the mission an organisation is implementing plays an important role in trying to determine what success will mean for the organisation. This mission usually reveals what will constitute success for a given organisation. "A well-defined and recognizable mission statement and the synergy resulting from such a mission statement" were among the critical success factors, determining the success of an organisation, pointed out in the study conducted by A.M. Travaille and P.H.J. Hendriks [12] with reference to an institute as part of a university.

If we assume that success means achieving set objectives, then with regard to a university, may it be assumed that one of the objectives of a public university is a profit/competitive advantage?

Since public universities as business entities report profits/losses (an excess, or not, of revenue over expenses) in their financial statements, then may we measure their success or failure using profits/losses and is this reasonable, as is the case for enterprises?

Conducting didactic and research activity, universities "sell" knowledge (for example, about competences, increasing resilience to crises [13]), educational services, and patents and take part in national and international projects, which generate specific benefits, including financial ones (that is, profits). As R. Plummer et al. claimed, in New Public Management, "higher education institutions are expected to account for their performance" [2]. Therefore, it seems justified to measure their success in terms of a clear/hard/measurable criterion such as profit and efficiency.

As in the case of success, establishing the key success factors for an organisation may be problematic. With the help of the literature on this subject, it may be stated that the key success factors (KSFs) are: strategic activities [14,15]; and important facts; skills or resources, unique characteristics of an enterprise, a heuristic tool to sharpen thinking [16], one of the key strategic tools [17].

According to the authors, the key success factors may refer to the most-important ones for achieving the objectives and delivering the mission and vision of external and internal unique characteristics/areas of operations, similar for organisations within one industry, which, when managed appropriately, lead to efficient competitive activity and translate into the competitive advantage of an organisation.

With regard to universities, key success factors are indicators of a competitive advantage that reflects the external value of a university [18].

The main success factors of any organisation, such as financial stability, the optimization of internal processes, and staff development [3], are complemented in the case of universities by the ones that are specific for their activity.

Assuming that the key success factors of universities are the areas/criteria considered and evaluated in the international and national rankings of universities, these may include: the quality of education, the quality of faculty, research output, per capita performance (Academic Ranking of World Universities (ARWU)); teaching, research, citations, industry income and international outlook (The Times World University Ranking (THE)); teaching and learning, research, international orientation, regional engagement, knowledge transfer (U-Multirank); visibility, transparency (openness), excellence (scholar) (Webometrics); prestige, graduates in the labour market, innovation, academic potential, academic effectiveness, teaching and learning, internationalisation (The Perspektywy University Ranking (Perspektywy)).

There is a certain convergence in the areas/criteria of university evaluation in most of the rankings mentioned, but also, some internal differences in the interpretation of their content are visible. These areas/criteria of HEIs' functioning, with some internal differences that can be seen in the interpretation of their content (indicators describing them), were basically common for all HEI rankings included in the study (teaching, research, internationalisation).

However, several additional areas specific to each ranking were highlighted, such as: performance per capita (ARWU), citations and industry revenue (THE), regional engagement and knowledge transfer (U-Multirank), visibility (Webometrics), or academic potential, innovation, and prestige (Perspektywy). For the purpose of this study, it was assumed that we may also identify and define the critical success factors (CSFs).

The literature on the subject offers various definitions of critical success factors, which are treated on a par with key success factors by many authors [19]. Nonetheless, we may also find statements that allow us to differentiate between KSFs and CSFs. Considering the most-important differences in the definitions of CSFs, it may be claimed that these are: a limited number of areas where the results, if satisfactory, guarantee a competitive

success to the organisation [3,20–24]; factors within the organisation, things that must be done for the organisation to succeed [25]; a limited number (usually from 3 to 8) of features, conditions, or variables [26]; elements that are necessary for the effective strategy's implementation [27]; physical location, security, and reliability [28].

Consequently, the authors consider critical success factors as a limited number (usually from 3 to 8) of internal features of an organisation, variables or factors indispensable for effective strategy implementation, typical of a given industry, which are necessary for an organisation to employ its vision and mission. The management should focus on these critical success factors as they enable the organisation to achieve competitive success in the market.

The above-mentioned definitions of the KSFs and CSFs may also be presented as follows: (1) KSFs—20% of internal and external factors (resources, skills, etc.) that affect the success of an organisation/university to the largest extent—in line with the Pareto principle; (2) CSFs—a limited number (3–8) of the most-important internal KSFs, without which the organisation/university cannot succeed.

This agrees with Nucińska's statement that "public entities need more effective management than commercial entities", which is due to the fact of "the lack of market incentives acting on them that would spontaneously force effective and efficient operation" [29].

In this context, a question arises: How may performance and efficiency be measured? Authors believe that this can be achieved through performance indicators (KPIs and CPIs) and efficiency indicators. An issue tightly linked to the study of key/critical success factors is their measurement, according to R.S. Kaplan and D.P. Norton's view: what you measure is what you get. Thus, a question arises of how to measure the level of KSFs/CSFs.

In the management literature and practice, we use the so-called key performance indicators (KPIs), which, according to J. Reh, are "values used to monitor and measure efficiency"; while some of them (e.g., net profit margin) are almost universal for various types of enterprises, "most industries also have their own Key Performance Indicators" [5]. KPIs are efficiency indicators that may be measured and analysed, which is why they are also used to create reference points and measure competition. KPIs are not objectives themselves; they are tools that we need to understand and measure success [6]. The definition of KPIs formulated by D. Parmenter suits the aims of this article perfectly, emphasising that these are "financial and non-financial indicators used as measures in the organisation performance measurement process" [7].

For the needs of this study, the authors identified KPIs and critical performance indicators (CPIs) with reference to universities: (1) KPIs—measures of various types referring to KSFs—qualitative and quantitative measures (also those collecting students' opinions) from various ranking lists; (2) CPIs—(in the opinion of university managements) key (objective/measurable/quantitative) measures that enable the analysis of the CSF levels.

Unlike corporations, which can measure success in cumulative share price, research universities require a range of qualitative and quantitative output measures. These include indicators such as: accreditation, international and regional rankings, research grants, research impact measures, teaching and learning metrics, financial audits, graduate outcomes, number of student applications, especially at the graduate level, and preferred employer status and level of philanthropic activity, among many other measures [30–32].

As a consequence of adopting the areas/criteria for the evaluation of HEIs included in the rankings as key success factors, the indicators enabling their measurement may be considered key performance indicators. Thus, key performance indicators for universities may be: alumni of an institution winning Nobel Prizes and Fields Medals, highly cited researchers (ARWU), staff-to-students ratio, papers-to-academic staff ratio, income-to-academic-staff ratio (THE), graduating on time (masters), citation rate, student mobility (U-Multirank), web contents' impact, top cited papers (Webometrics), international recognition, parametric evaluation, and accreditations (Perspektywy), just to mention a few.

The second question is: How is efficiency measured? In the context of the Web of Science or Scopus literature review, the authors noticed an incentive to focus on the

technical efficiency to estimate the level of efficiency in public higher education institutions. Technical efficiency is understood as the ability of a production unit to secure the maximum volume of outputs using the given number of inputs, respectively to use the minimum volume of inputs in order to produce the given number of outputs [33]. On the other hand, Mikusova, based on his own literature research, stated that efficiency—the average efficiency score of public universities—is measured by the arithmetic mean. The authors agree with Johnes and Ruggiero [34] that the assessment of the effectiveness has many different variants. The authors note that technical efficiency in higher education can be measured by non-parametric (e.g., data envelopment analysis (DEA)) or parametric (e.g., stochastic frontier analysis (SFA)) methods.

According to the literature review, it can be noted that many authors measured the efficiency of higher education institutions using the DEA method [35–45], while others used the SFA method [38,46,47]. In Wolszczak and Derlatka's opinion [48], the analysis of education institutions' productivity is different from standard productivity measurements, not only because no profit is maximised here, but also because HEIs are not standard firms with one output and a set of inputs. On the contrary, HEIs are producers of at least two outputs: teaching and research. The methodology of efficiency measurement has to take this specificity into account. Because the authors agree with this opinion, they propose to research the efficiency of Polish HEIs by indicators that are used at companies.

Godínez-Reyes et al. [49] argued that corporate efficiency is given by three indicators: return on assets (ROA), return on equity (ROE), and return on sales (ROS). Droj et al. claimed that the most-common financial indicators are: ROE, ROA, solvency, and ROS [50]. In the context of meta-analyses, one the efficiency of public HEIs, Mikusova [33] claimed that "efficiency" means the average efficiency score of public higher education institutions.

An indicator that is identified with the measurement of efficient asset management is the return on assets. Thus, the authors decided that it would be applied to measure the efficiency of the functioning of public universities in Poland, and more specifically, it would be the ROA, measuring the asset profitability ratio and assessing the rate of return on assets (which shows to what extent the PLN worth of assets contributes to generating of financial result), calculated according to the following formula:

$$\text{ROA} = (\text{net profit/total assets}) \times 100\%$$

Another indicator for assessing asset profitability may be the return on equity (ROE) or the return on investment (ROI). The indicator ROE is calculated according to the following formula:

$$\text{ROE} = (\text{net profit/equity capital}) \times 100\%$$

However, due to the lack of data, only ROA and ROE were measured for all 10 indicated universities. Unfortunately, the main obstacle was the lack of data, especially in economic terms, for Polish public universities. The authors agree with Wolszczak and Derlatka [48] about improving the policy of microdata collection and dissemination at the European level, but also call for a more transparent microdata policy at the national level.

Universities in Poland—based on the Act on Public Finance [51] and other acts regulating their operations, such as the Law on Higher Education, the Public Finance Act, and the Regulation of the Council of Ministers on principles for public universities' finance management—keep financial and accounting records, make financial data available, as well as draw up public statistics with regard to ministry grants. However, as U. Teichler pointed out, owing to the lack of uniformity of the used terms or concepts, which refer to the phenomena, and owing to a composite approach to funding, the lack of the comparability of data referring to public universities may be observed, which inhibits the collection of interesting information and also limits the verification of the accuracy of the data adopted for the analysis [52].

Generally, the analysis of the public finances of higher education institutions may resemble the financial analysis of an enterprise. The review of methods for financial analysis should be aimed at the selection of indicators adequate for the analysis of the functioning of a university.

In line with Article 28 of the Act on Public Finance, it is recommended that "a budget policy institution manages its assets on its own, guided by the principle of efficiency of their use" [51]. Pursuant to this guidance, the management of a public university should pay attention to its finances and make an effort to efficiently manage the assets owned.

As noted by Brzezicki et al. [35], already in the assumptions of the Act on Higher Education of 2011, it was indicated that "the proposed changes are primarily pro-quality and lead to an improvement in the efficiency of spending public funds on higher education". These authors [35] rightly pointed out that "by 2030, development goals were adopted, including improving the quality of education (...), improving the functioning of the higher education system through changes in the areas of organisation, management and financing of teaching activities".

In the context of university management, efficiency is considered in terms of good governance, and it refers to the further application of the results of the conducted measurements to make comparisons between universities. Wilkin [53] distinguished the following: ensuring adequacy, which means whether a selected indicator is appropriate for the specificity of university operations; the ability to perform measurements and specify a standard by which a given feature will be measured; gaining access to the data needed to make a measurement; the ability to acquire data not only for the period when the measurement is performed, but also the ability to acquire a time series of data, while information should be gathered and generated by universities cyclically; drawing attention to the possibility of obtaining comparability not only between universities in a given country, but also internationally.

## 3. Materials and Methods

The research procedure consisted of five main stages, which helped to answer three research questions. The first stage was the selection of universities for the research, which included the identification and analysis of international and Polish ranking lists of higher education institutions (HEIs) on which Polish HEIs were included (ARWU, THE World University Ranking, U-Multirank, Webometrics, Perspektywy), then the lists of public universities in Poland for further analyses were prepared. The lists consisted of universities that were ranked in at least four ranking lists published in 2019 (17 universities). The final stage in this part of the research was to create a database of the addresses of particular groups of experts. The second stage of the research procedure included the preparation of the interview questionnaire, which was sent in an electronic form to expert, of 17 universities from 8 May to 10 June 2020. Thanks to this, there were 53 experts' opinions on the identification of critical success factors for public universities and of critical performance indicators, which resulted in singling out three critical success factors (CFSs) and eight critical performance indicators (CPIs).

To answer the first research question, the authors applied an expert opinion research method, using a structured interview questionnaire, which was sent by email. The questionnaire consisted of 6 demographic questions (semi-open multiple-choice and close-ended single-choice questions) and 10 factual questions on critical success factors. This section included semi-open multiple-choice questions and questions on a 10-point semantic differential scale. This method was selected to obtain opinions due to the COVID-19 pandemic situation. As experts, the authors considered persons employed in top managerial positions at the universities: rectors and vice-rectors, chancellors, and deans.

The next stage of the research procedure consisted of selecting the group of public universities in Poland out of 59 public universities supervised by the Ministry of Higher Education and Science, which met the following conditions: were ranked from 1 to 50 in the Perspektywy national ranking list in 2019 [54]; published a profit and loss account

including the "Profit/Loss" item as of 31 December 2018; published profits/losses as of 31 December 2018, and not their increase/decrease as compared to the previous year. As a result, for further analysis, 10 universities were selected.

To answer the second research question, the financial documents of public universities in Poland were used and analysed. The last stage was a comparison of previously identified CPIs and the ROA/ROE ratios of 10 selected HEIs on the matrix of the success level of HEIs in Poland created for this purpose.

To answer the third research question, the authors' own actuarial matrix of the success level, which juxtaposes the identified CPIs and the ROA/ROE, was used to measure the efficiency of public university operations.

## 4. Results

The higher education system in Poland comprises public and non-public institutions. Within the 133 public higher education sector's institutions, we may distinguish vocational schools and various types of universities supervised by public institutions (ministries) and church institutions. Public universities, whose number has remained stable over the years, are supervised by the Ministry of Education and Science (formerly the Ministry of Science and Higher Education), the Ministry of Culture and National Heritage, the Ministry of Health, the Ministry of National Defence, the Ministry of Internal Affairs and Administration, the Ministry of Marine Economy and Inland Navigation, and the Ministry of Justice.

From 2012 to 2018, the number of higher education institutions in Poland decreased from 456 to 392 entities. The largest share in this decrease may be attributed to non-public higher education institutions; since 2012, their number has dropped by 63. However, the number of public and church-related institutions has remained stable [55].

In this study, the authors focused on public higher education institutions supervised by the Ministry of Education and Science; in 2018/2019, there were 59 such institutions. The number included 18 universities, 18 technical universities, 5 economic universities, 5 pedagogic universities, 6 agricultural/natural sciences universities, 6 schools of physical education, and 1 theological institution [56].

A structured interview questionnaire was filled in by 53 experts including 13 rectors and vice-rectors, 5 chancellors, and 35 deans of the selected HEIs. The experts were 8 women and 44 men. One person preferred not to specify gender. Eighteen respondents declared a background in engineering and technology, thirteen a background in social sciences, thirteen in science and natural sciences, four respondents in humanities, four in agriculture, and one person in both social sciences and humanities. When asked about the length of service, 50 respondents answered that they had been working for more than 20 years and 3 respondents said they had been working for between 10 and 20 years. In terms of the total length of service in university management positions, 15 respondents declared more than 20 years, 20 experts between 10 and 20 years, and 18 experts less than 10 years.

### 4.1. Identification of Critical Success Factors and Critical Performance Indicators for the Public University Sector in Poland

The authors developed a model for identifying critical success factors and critical performance indicators, as shown in Figure 1.

The model for the identification of the CSFs and CPIs included the following components:

Component 1. The KSFs were identified based on the analysis of national and international ranking lists of higher education institutions and the analysis of the literature on the subject.

Figure 2 presents the results of the research aimed at the identification of the key success factors (Component 1 of the model in Figure 1).

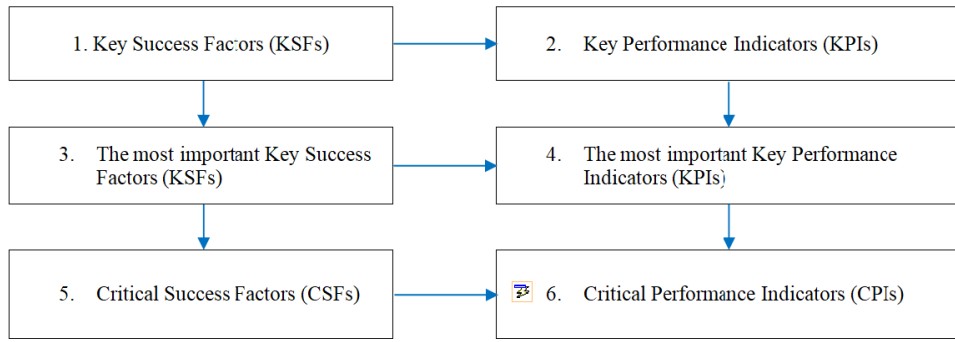

**Figure 1.** The model for the identification of the critical success factors (CSFs) and critical performance indicators (CPIs).

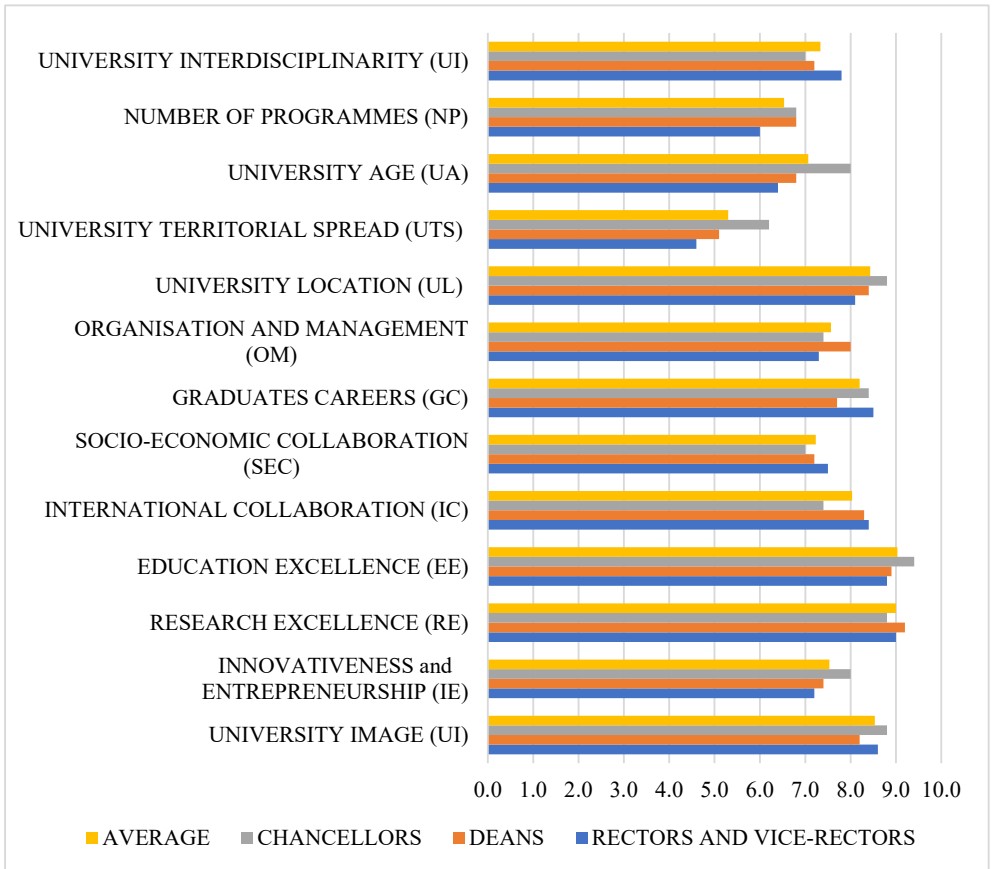

**Figure 2.** Key success factors according to experts—internal stakeholders holding top positions in the university hierarchy (rectors, vice-rectors, deans, chancellors).

Component 2. The KPIs for individual KSFs—identified based on the analysis of national and international ranking lists of higher education institutions and the analysis of the literature on the subject (due to the article's space limitations, the authors decided not to indicate the 68 KPIs included in the research).

Component 3. The most-important KSFs identified based on the experts' opinions regarding the average weight and dominant individual KSFs (with the highest average weight from 8.0 to 10.0 and dominant from 8 to 10)—based on the structured interview conducted using Google Forms and processed with the Microsoft Excel 2007 spreadsheet software.

With regard to Component 3 of the model (Figure 1) and taking into consideration the experts' opinions, it may be concluded that persons holding top positions in the hierarchy

of the analysed universities, who were deemed experts in the conducted study, saw the following as the most-important key success factors, as shown in Table 1.

**Table 1.** The most-important key success factors in the experts' opinion.

| Key Success Factors | Average Weight | Dominant |
|---|---|---|
| Research Excellence (RE) | 9.0 | 10 |
| Excellence in Education (EE) | 9.0 | 10 |
| University Image (UI) | 8.5 | 9 |
| Location of the University (UL) | 8.4 | 10 |
| Careers of Graduates (GCs) | 8.2 | 8 |
| International Cooperation (IC) | 8.0 | 9 |

Component 4. The most-important KSFs identified based on the experts' opinions regarding the weight of individual key success factors and the dominant ones (KSFs with the highest average weight from 8.0 to 10.0 and dominant from 8 to 10)—based on the structured interview conducted using Google Forms and processed with the Microsoft Excel spreadsheet software.

Considering Component 4 of the model for identification presented in Figure 1, the authors singled out the 15 most-important KPIs, which are pictured in Figure 3.

Component 5. CSFs—including the identified most-important KSFs in line with the adopted definition of the CSFs.

With regard to Component 5 of the model and with reference to the previously formulated definition of the critical success factors, this study adopted the following as the CSFs: research excellence, education excellence, international cooperation.

The identification of the above-mentioned critical success factors allowed the authors to focus on further research solely on 3 CSFs so as to specify the most-important measures that enable their quantification (CPIs). This is the ultimate element of the model in Figure 1.

Component 6. CPIs within 3 CSFs—the CPIs were the KPIs that were indicated most frequently—depending on the CSF, from 50 to 100%.

Having taken into consideration the experts' opinions, the following eight CPIs were identified for individual CSFs: (1) research excellence: number of citations of university employees' publications in international databases; the sum of parametric grades granted by the Ministry of Science and Higher Education to university units; the number of publications in international databases; (2) education excellence: the number of current accreditations and international certificates and Polish Accreditation Committee (PKA) accreditations with an excellent grade held by the university; (3) international cooperation: the number of foreign students as compared to the total number of students; the percentage of international joint scientific publications of universities; the number of students participating in foreign exchange programmes (at least 3 months/one semester), as compared to the total number of students.

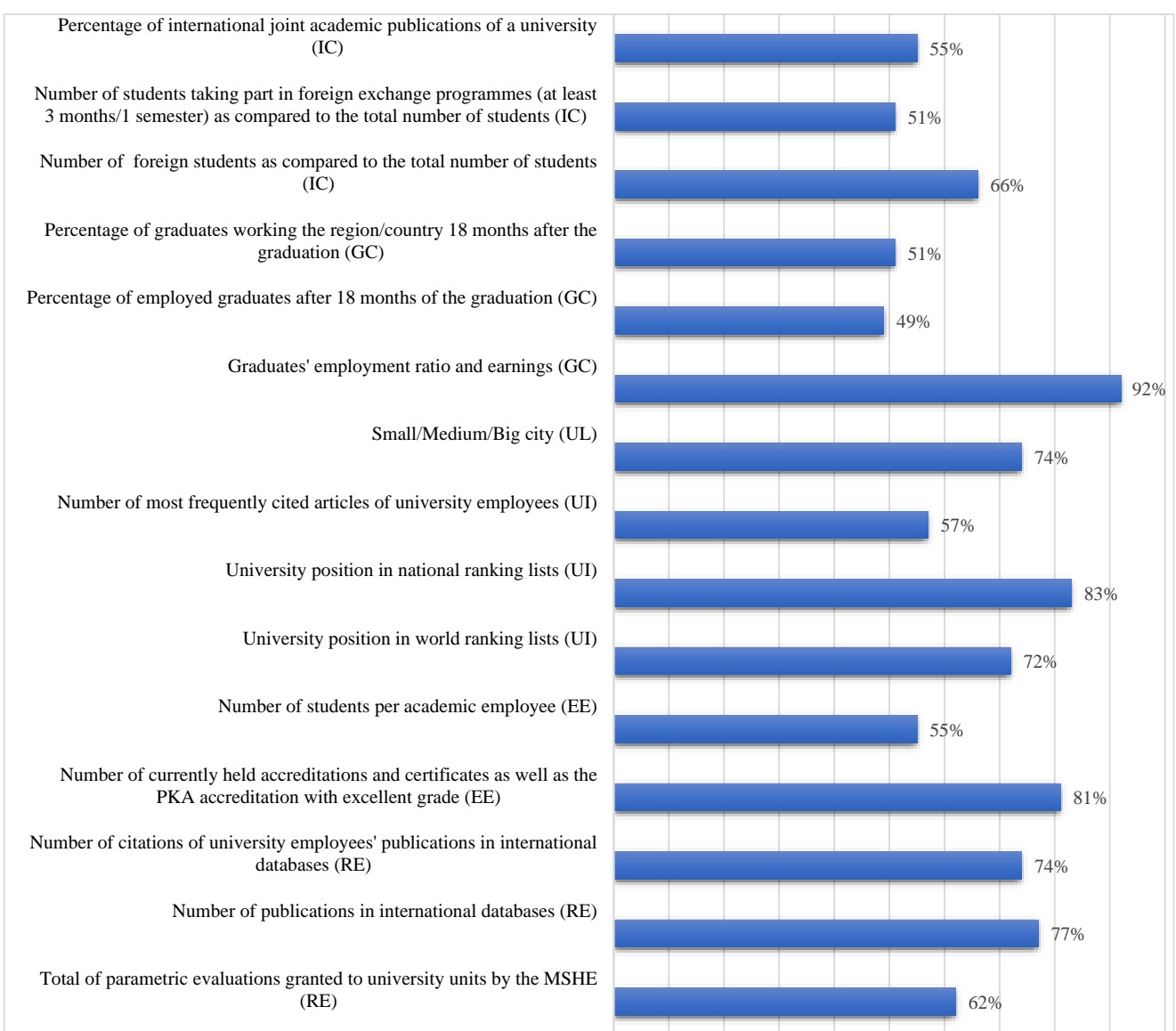

**Figure 3.** The most important key success factors and key performance indicators according to the experts—internal stakeholders holding top positions in the university hierarchy (rectors, vice-rectors, deans, chancellors).

### 4.2. List of Efficiency (ROA/ ROE) and Performance Indicators (CPI) of Selected Public Universities in Poland

To assess the functioning of public universities from the good governance perspective, the authors analysed the level of efficiency using an economic and financial indicator used in economic practice, i.e., ROA/ROE [57].

The analysis of the efficiency was conducted on the sample of 10 public universities, which included 5 universities and 5 technical universities (U1: Jagiellonian University, U2: University of Warsaw, U3: Warsaw University of Technology, U4: University of Lodz, U5: Nicolaus Copernicus University, U6: Adam Mickiewicz University, Poznań, U7: AGH, U8: Poznań University of Technology, U9: Wroclaw University of Science and Technology, U10: Warsaw University of Life Sciences).

The results of the ROE and ROE comparison are shown in Figure 4.

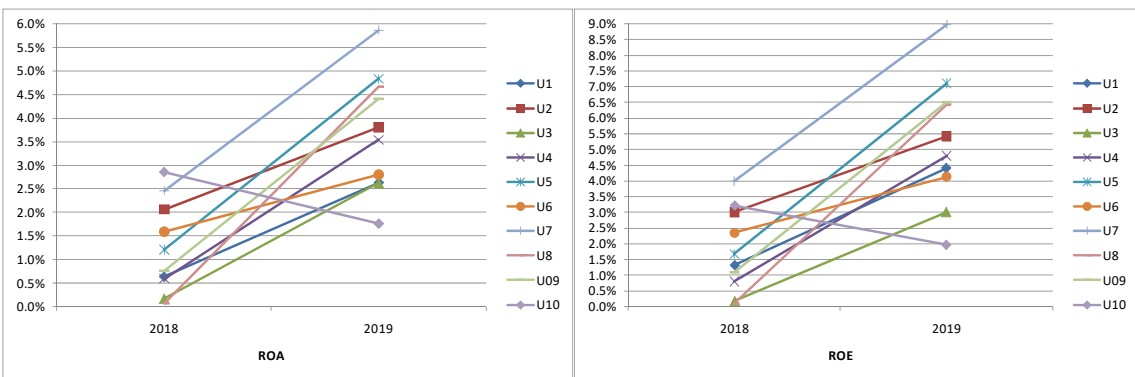

**Figure 4.** ROA and ROE for selected public universities in Poland.

In the analysed period, the universities obtained positive ROA values, and in the case of nine units, ROA increases were recorded. In 2018, the highest increases—achieved above the average ROA (1.24%)—were achieved by 4 universities (U10, U7, U2, and U6), while in 2019, 5 universities (U7, U5, U8, U9, U2) achieved an above-average ROA (3.69%).

Based on the ROE data in the period considered, it can be seen that the indicator took positive values. Changes in the ROE value over time indicated an improvement in the situation in 9 out of 10 universities surveyed. In this context, the best positions—above the average ROE (1.78%) in 2018—were achieved by four universities (U7, U10, U2, U6), and a year later, 50% of the units (U7, U5, U9, U8, U2) obtained an above-average ROE (5.28%).

Considering the data in Figure 4, it should be noted that U10 (Warsaw University of Life Sciences) did not follow the trend due to the declining ROA and ROE associated with the declining net profit in the years under review.

In the case of the performance indicators from every CSF, one CPI, which was indicated the most frequently by the experts, was adopted for further analysis [58]: (1) research excellence: the number of publications included in the Scopus database in the years 2016–2020, as compared to the total amount of research, as well as research and didactic employees; (2) education excellence: the number of current accreditations and international certificates, valid PKA accreditations with an excellent grade (awarded until 2018), and education excellence certificates (awarded currently); (3) international cooperation: the number of foreign students as compared to the total number of students.

For each CPI, the average value was calculated, which was the position of the matrix division on the OX axis (Point a in Figures 5–7), and so, for:

- The number of publications of university employees in international databases, the average was 62.53;
- The number of accreditations and international certificates held by the university, including excellent grade accreditations awarded by the PKA, the average was 52.38;
- The number of foreign students as compared to the total number of students, the average was 7.22.

In the case of the OY axis, the division point was the average value of ROA (Point b = 3.69) or the average value of ROE (Point b = 5.28) (Figures 5–7).

The criterion taken into consideration when selecting universities that showed high efficiency was the level exceeding the average ROA for universities comprising the research sample > 3.69%and the level exceeding the average ROE > 5.28%.

In the area of research excellence (RE), the CPI analysis was conducted—considering the number of publications of university employees in international databases. The source data came from SciVal, a licensed bibliometric tool for the analysis of data included in the Scopus database, which allows for the assessment of one's own and others' research activity from various perspectives so as to help prepare, execute, and assess strategies based on reliable evidence [59].

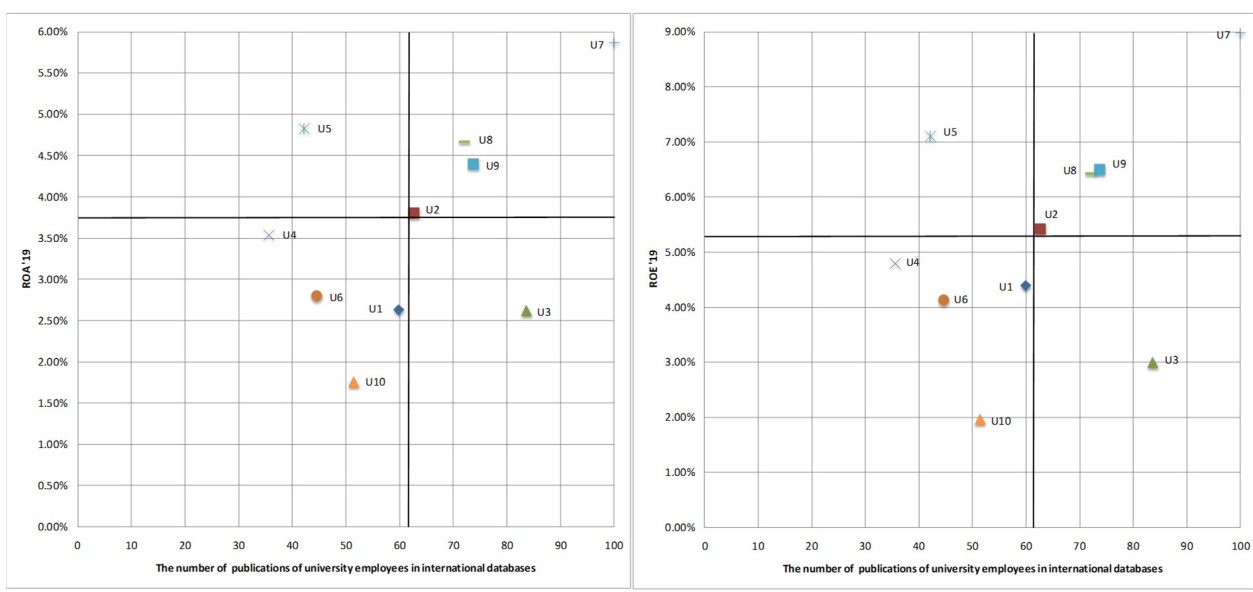

**Figure 5.** ROA and ROE vs. the number of publications of university employees in international databases.

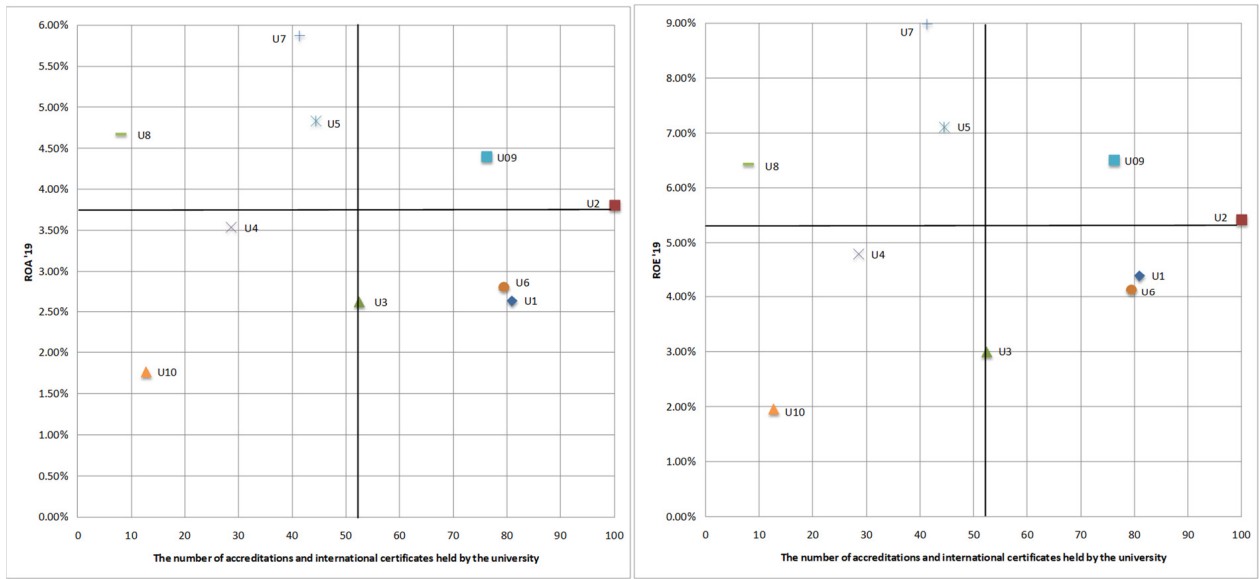

**Figure 6.** ROA and ROE vs. the number of accreditations and international certificates held by the university, including excellent grade accreditations awarded by the PKA.

Based on the data shown in Figure 5, it may be stated that, in the research sample, the highest level of effectiveness measured, on the one hand, with ROA and ROE exceeding the average, the ROA and ROE value for sampled universities, equal to 3.69 and 5.28% in 2019, and, on the other, the level of CPI measured with the number of publications exceeding the average (62.53), was recorded for universities U7, U8, U9, and U2. Four universities (U10, U1, U6, U4) took relatively the least-favourable positions.

In the area of education excellence (EE), experts indicated the number of accreditations and international certificates as the most-important key performance indicator for public universities in Poland (Figure 6). The source for the data was the PKA database and the databases of international accreditation agencies [5].

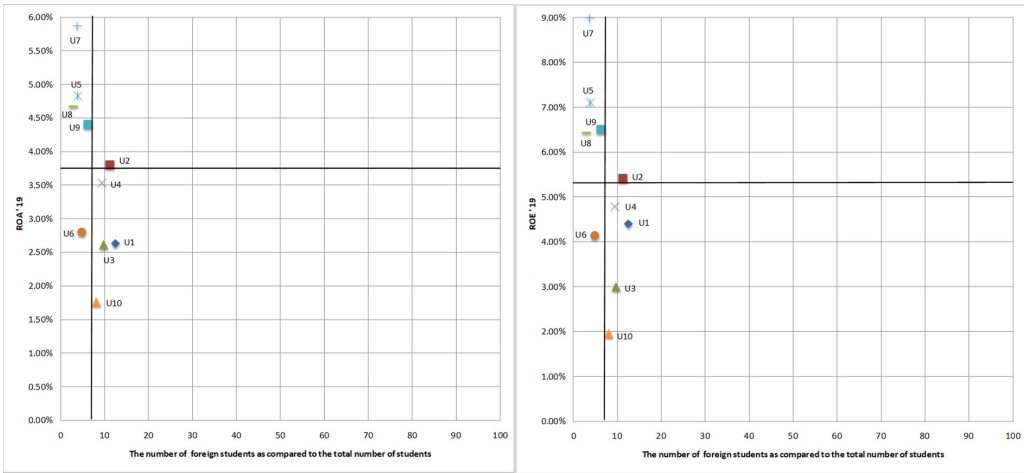

**Figure 7.** ROA and ROE vs. the number of foreign students as compared to the total number of students.

As shown by the data in Figure 6, three universities, U2, U5, and U9, were in the quarter with a high level of effectiveness and a large number of accreditations (>52.38). U4 and U10 performed relatively the worst.

In the area of international cooperation (IC), experts indicated the number of foreign students as compared to the total number of students as the most-important key performance indicator for public universities in Poland (Figure 7). The source of the data was POL-on.

As shown by the data in Figure 7, only one university—U2—was in the quarter with a high level of effectiveness and a large number of foreign students (with an average >7.22), and one university (U6) was in the quarter with a low level of effectiveness and a low level of the CPI.

When referring to the CPI, such as the number of foreign students in relation to the total number of students, it should be noted that, in general, the majority of Polish public universities were relatively poorly advanced in this respect.

The research results illustrated in Figures 5–7 show that the highest level of effectiveness was achieved successively by the following public universities in Poland: U2, U7, U8, and U9.

## 5. Discussion and Conclusions

As a result, it should be emphasised again that success may be and should be considered in many dimensions with regard to various types of organisations, including universities, which constituted the main subject of the analysis in this study. In the case of higher education institutions, success exists at different levels, as shown in Figure 8.

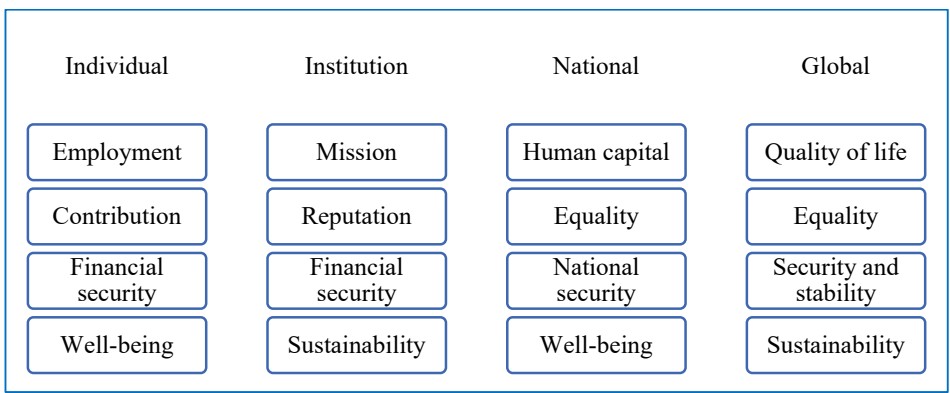

**Figure 8.** Success factors for university stakeholders. Source: [60].

Taking into consideration the nature of higher education institutions, their success may also be identified with: (1) the development of the city/region/country where the university operates, acting on the assumption that the more people the university educates, the richer/more innovative the country/region/city is; (2) the job satisfaction and employability of university graduates [61]; (3) the employer's satisfaction from having educated personnel, which translates into the higher profits of the enterprises where they are employed; (4) the number of graduates employed in the city/region/country; (5) a large interest from foreign universities in cooperation in the scope of didactics, as well as research; (6) the adoption of an appropriate strategy [62]; (7) in the context of academic entrepreneurship, as the constant and continuous generation of income, both for universities, as well as for industry partners [63]; (8) or as a profit.

The aim of the article was to highlight the factors by which it is possible to measure the success of public universities and the levels of success that can be achieved in the HEI sector. The goal was achieved because the factors that measure the university's success were identified, and the following part of the conclusions will present the levels of success resulting from Figures 5–7.

The main research problem in the form of the question, "What are the measures and levels of success of public universities?", was divided into three specific research questions:

RQ1: Which key success factors can be considered critical to the success of public universities?

Having taken into consideration the experts' opinions, which allowed identifying the critical success factors (Figure 1), and with reference to the previously formulated definition of the critical success factors, this study adopted the following as the CSFs: research excellence, education excellence, international cooperation.

RQ2: Can performance (CPI) and efficiency (ROA, ROE) indicators be considered as measures of success of public universities?

Having taken into consideration the experts' opinions, which allowed identifying the critical performance indicators (Figure 1), and with reference to the previously formulated definition of the critical performance indicators, this study adopted the following as the CPIs: the number of publications included in the Scopus database in the years 2016–2020, as compared to the total amount of research, as well as research and didactic employees; the number of current accreditations and international certificates, valid PKA accreditations with an excellent grade (awarded until 2018), and education excellence certificates (awarded currently); the number of foreign students as compared to the total number of students.

As a result of the research, it should be stated that, in the analysed period, in the case of the CPI, which was the number of publications of university employees in international databases, and taking into account the average of this indicator for the analysed universities (62.53), it should be stated that five out of the ten o selected universities (U2, U3, U7, U8, and U9) achieved a higher level than the average. In the case of the number of accreditations and international certificates held by the university, including excellent grade accreditations awarded by the PKA, with the average 52.38, four universities (U1, U2, U6, and U9) achieved a higher level than the average and one university (U3) achieved the average level. Finally, in the case of the number of foreign students as compared to the total number of students, with an average of 7.22, five universities (U1, U2, U3, U4, and U10) achieved a higher level than the average.

Taking into account the obtained results concerning the level of the CPIs of the selected universities and the general knowledge on universities in Poland, it should be stated that the analysed universities were the ones that achieved success, in whatever way it is understood. If the reputation or position in the rankings is taken into consideration as a success factory of the university, then Jagiellonian University, University of Warsaw, Warsaw University of Technology, and Wroclaw University of Science and Technology were public universities that appeared in various international rankings and were at the forefront of Polish public universities.

The authors are aware of the fact that the CSFs and CPIs they proposed are not always, and not for all university stakeholders, important. For university leaders, there may be a completely different list of factors and indicators (not always parametric) than for employers or students.

However, it is important that such factors be identified and measured, also from different points of view and in a dynamic manner so that the university can achieve competitive success in the market.

When answering the question about whether the efficiency indicators (ROA, ROE) can be considered as measures of the success of public universities, the authors claim that this is in line with the assumptions of New Public Management and the Positioning School of Strategy.

New Public Management is characterised by the use of markets (and quasi-markets) that drive competition between public sector suppliers; empowered entrepreneurial management; clear standards, measures of performance, goal setting, and quality assurance mechanisms; and a focus on results [64,65]. However, it must be said that New Public Management may have negative consequences for the functioning of universities, as pointed out by C. Ross et al. (2022). Nonetheless, in an era of quantifiability, measurability, and productivity, the research on university effectiveness is one of the most-important international trends in public administration [66].

Referring to the Positioning School of Strategy, it assumes that profit may be considered the quantitative measure of a company's success [1]. Surely, this approach to success is mostly attributed to enterprises, with one of their key objectives being profit maximisation in the long run. Profit, as a measurable economic category, allows determining in various periods of organisation functioning whether a given organisation achieves its objectives—whether it succeeds. Nonetheless, as Bailom, Matzler, and Tschemernjak [58] pointed out, "success may not be defined solely in terms of financial indicators, as they are based on past achievements and are too late to register changes within and outside the organisation".

The profit of a university, as a measure of the success of public universities, is generated from many sources, which include, among other things, grants for students from the Ministry of Education and Science, sales of patents, fees for paid programmes, and university ancillary activity.

Certainly, profit maximisation is not the main objective of a public university, but successful universities do generate profit on their activity. However, as shown in the study by B. Uslu [67], the factors he analysed (citation, income, internationalisation, prize, publication, reputation, and ratios/degrees) were statistically significant; still, only one of them—i.e., income—was a negative factor.

As a result of the research carried out, it should be stated that, in the analysed period, the ROA values were positive, which means that all universities selected for the analysis generated profits, not losses. In the case of nine public universities, ROA increased, which proves the increasingly better financial condition of the surveyed units. The higher the ROA value, the better the financial condition of the university is, and in this context, U7 obtained the best position. In the analysed years, the highest increases—achieving an above average ROA = 1.24 (2018)—were achieved by 4 universities (U10, U7, U2, and U6), while an above average ROA = 3.69 (2019) was achieved by 5 universities (U7, U5, U8, U9, and U2).

On the basis of the ROE data in the analysed period, a favourable situation of the analysed universities can be noticed, because the indicator took positive values, which means that all units generated a profit, not a loss. This means that, in the case of the studied universities, an accounting surplus of revenues over costs was obtained, and the leader in this respect was U7. Changes in the value of ROE over time indicated an improvement in the situation in 9 out of the 10 universities surveyed. The stimulant ROE means that the higher its value, the more favourable the situation of the individual is. In this context, the best positions—an above average ROE = 1.78 (2018)—were achieved by 4 universities (U7,

U10, U2, U6), and an above average ROE = 5.28 (2019) was achieved by 5 universities (U7, U5, U9, U8, U2).

Considering the obtained results concerning the level of ROA/ROE of the selected HEIs in Poland, it should be stated that the analysed units were successful, as they generated a surplus of revenues over the costs, thus obtaining a positive financial result, which proved the effective management by the university leaders.

The authors are aware that the efficiency of HEIs can be measured by other indicators (besides ROA/ROE, for example ROI—return on investment, ROS—return on sales), also in a dynamic way to verify whether the HEI is managing assets efficiently and succeeding in the HEI market.

RQ3: What are the levels of success of the selected public universities in Poland?

A comparative analysis was conducted of the selected CPIs (indicated by experts as the most-important ones) and ROA/ROE, which may be used to measure the efficiency of the functioning of public universities in Poland. The results are presented in charts, where the ordinates (OY) show the value of ROA/ROE and the abscissae (OX) the CPI value. The values are given in ascending order; the further from the origin of the axis, the better the results achieved by the university. The point of division of the CPI and ROA/ROE levels is the average values. The charts are divided into quarters, and the details are presented in the form of a matrix (Figure 9).

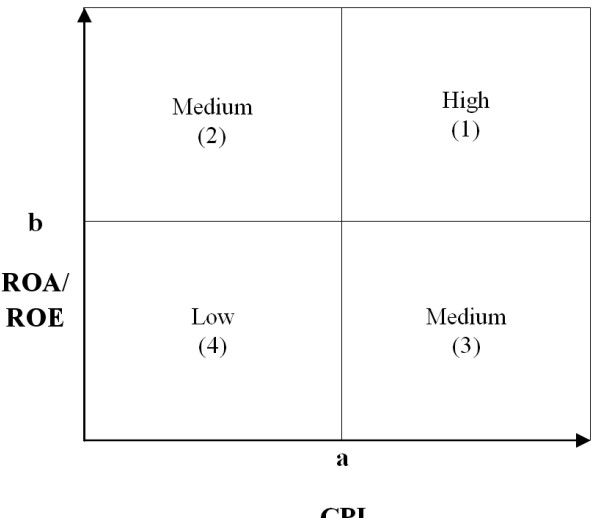

**Figure 9.** Matrix of the success level.

The authors assumed that universities may reach three levels of success:

(1) High/maintain the level: A strategic recommendation for university authorities is to maintain a satisfactory level of the performance indicators (e.g., ROA, ROE above the average in the sector) and care about the CPI in order to maintain a strong competitive position and achieve success in the market as a leader;

(2,3) Medium/monitor: The recommendation is to monitor and make greater efforts to increase the level of the performance indicators (e.g., ROA, ROE) and a sensible approach to analyse each CPI to increase the university's success and importance in the market;

(4) Low/keep informed: The "urgent" and "important" recommendation is to increase engagement and keep informed of progress on the dynamics of the performance indicators (e.g., ROA, ROE) and to devote significant attention to achieving more satisfactory levels of the critical factors that determine success in the sector.

Meeting the assumptions of good governance, the tool created for the purpose of this study in the form of the matrix of the success level (Figure 9) may be used for the analysis of this level both statically and dynamically, showing the manner in which the level changes over time and/or how it may/should change. In addition, it can be used to

analyse the success of HEIs taking into account other variable criteria, e.g., the juxtaposition of financial indicators with environmental or social criteria, and it can take into account relations with internal and external stakeholders. As a result, it can visualise the fact of the achievement/scalability of the sustainability policy objectives. Thus, it can be used as a tool to improve activities in key areas of HEIs' functioning by those in managerial positions, and at the same time, it can contribute to more-effective sustainable management of public HEIs.

It seems that public HEIs are "financed from public funds, their activities and results are in the public interest" and that HEIs should have "academic self-accountability". However, the emphasis is now rather on "stronger external monitoring and evaluation processes and public accountability" [68].

Owing to the fact that, on the one hand, public universities in Poland are funded in the form of targeted grants by the state, they need to submit financial statements of their operations to competent ministries in a form dedicated to the sector of public universities and they need to manage the allotted funds appropriately. Nonetheless, they also conduct service operations that generate additional revenue, which translates into their profits.

On the other hand, apart from accounting for funds, public universities need to implement the targets of adopted strategies, which are in line with the general strategy of higher education in Poland. The development strategies of universities need to attain objectives in the area of research, teaching, and socio-economic cooperation.

The results of the study conducted in the form of an interview questionnaire for persons holding top managerial positions in universities showed that these are the areas that were deemed the most important, that is the key success factors. This means that what experts consider a success was achieving objectives included in university strategies in the context of tasks and the mission of higher education in Poland (in line with the assumptions of the Planning School of Strategy).

The authors are aware that measuring the success of public universities only by performance and efficiency, which are the values of New Public Management, is not sufficient in light of the broader goals that universities should pursue. As Broucker et al. claimed, "These broader goals are of particular importance to higher education as a policy sector" [69].

However, given the dynamics of the change in the socio-economic environment associated with technological advances enabling more-effective parameterisation in terms of cost-effectiveness, the proposed indicators may, on the one hand, facilitate the building of partnerships with labour market institutions and, on the other hand, offer some guarantee of a high level of educational provision. It is envisaged that reliable analysis (e.g., on the basis of indicators) will be the basis for consumer decisions (e.g., a student interested in an educational offer or an entrepreneur wishing to benefit from research and development work) in the new digital economy.

In addition, it should be emphasized that, currently, one of the important goals of universities is their sustainable development, considered in two contexts: internal and external. Internally-driven sustainability of the university is dominated by the paradigm of the developing economy and also forces public universities to manage themselves economically, to be rational in the use of resources, to have a positive financial result, to be able to generate profit, to be economical, and in line with the idea of sustainable development, to stop the drive towards the wasteful use of resources. Therefore, it should consider the value of the university measured in economic terms (its productivity and efficiency), but also by its impact on internal stakeholders (students, university staff). Externally driven sustainability is determined by the relationships and a balanced dialogue with external stakeholders, as initiatives for the benefit of the community are realised through cooperation between the university and the business communities. That is why it should take into account and measure the impact of the university on external stakeholders (e.g., employers, the local community, or more broadly, society in the regional/national/international/global dimension) and on the environment, using quantitative and qualitative parameters. These

elements can be a set of key/critical success factors for universities and should be included in their sustainable development strategies.

The ultimate conclusion, therefore, is that sustainability is not only about universities pursuing increasingly ambitious business goals that take into account economic values, but also taking into account non-economic values and the expectations of different stakeholder groups in order for the university to thrive. It is, therefore, about building a socially and environmentally responsible university.

**Author Contributions:** E.M.: conceptualisation, methodology, original draft preparation, formal analysis, investigation, visualisation, resources; M.W.-A.—conceptualisation, methodology, original draft preparation, formal analysis, investigation, visualisation, resources; B.S.: conceptualisation, validation, review and editing, resources; J.B.: conceptualisation, validation, review and editing, resources. All authors have read and agreed to the published version of the manuscript.

**Funding:** This research received no external funding.

**Institutional Review Board Statement:** Not applicable.

**Informed Consent Statement:** Not applicable.

**Data Availability Statement:** Not applicable.

**Conflicts of Interest:** The authors declare no conflict of interest.

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
