# Peer review of "Application of Performance and Efficiency Indicators in Measuring the Level of Success of Public Universities in Poland"

_sustainability, doi:10.3390/su151813673_

Round 1
Reviewer 1 Report
The article titled “Application of performance and efficiency indicators in measuring the level of success of public universities in Poland” does not fit the issues covered by the journal. However, the current content is correct, but for educational type magazines.
It is recommended to incorporate criteria related to sustainability and the environment. In the same way, it is recommended to incorporate a deep analysis of these issues, to know if they reach the level of success of public universities in Poland.
Author Response
We thank you for your comments and recommendations, however, the idea behind the authors of the study was to refer to one of the specific themes of the Sustainability Journal/Sustainable Products and Services thematic issue, namely 'sustainable policies' for measuring the success of universities. While the authors recognize the need to use parametric and non-parametric methods to fully assess the success of selected public HEIs, in this article they focused on measurable, parametric criteria such as efficiency indicators and measurable critical success factors. The performance indicators showed the good financial condition of the selected institutions. Critical success factors and critical performance indicators were defined and performance and efficiency measures were used to assess the success of Polish public HEIs. The proposed results-oriented measurement tool can help leaders of public HEIs in strategic decision-making, and maintaining a good level of economic efficiency demonstrates the achievement of the policy objectives of sustainable development of public HEIs in Poland.

Reviewer 2 Report
1. The literature review carried out quite systematically, but must added publications 2021-2022 to substantiate the author's approach. In addition, attention should paid to well-known an annual ranking of universities QS World University Rankings based on eight key indicators.
2. Recommendation: description method of processing of expert assessments should supplemented with information about the software environment in which the processing was perform. Also necessary to check the consistency of expert assessments for example, using the concordance coefficient.
3. The tables and schemes properly show the data of research. For to interpret and understand Figure 4 it is advisable to present in the form histogram.
4. Figures 5-7 are representations of the success rate matrices (one of the financial measures and one of the Critical Success Factors). However, this does not take into account the effect of the combined influence of all factors. In this case, it is more expedient to perform a cluster analysis of the studied universities and build a matrix of success levels, taking into account a set of factors.
Author Response
1) The authors have supplemented the literature review with publications from 2021-2022 (highlighted in yellow). Thank you for drawing attention to the QS World University Rankings. The authors are aware of this and other international university rankings (for example: EduRank, UniRank, CWST Leiden, CWUR) and considered them at the beginning of the research, but decided that using data from 4 international rankings and 1 national ranking in the article was sufficient for the purposes of the article.
2) In the content of the article, the authors have included information about the Microsoft Excel Spreadsheet Software used (yellow text). However, they would like to emphasize that the authors intended the experts' opinions to be rather a starting point for further analyses, and the statistical analysis used was the background for the research conducted, so the use of the concordance coefficient in relation to the experts' opinions was not the main intention of the article.
3) The aim of the authors of the reviewed article was to present trends and tendencies in the field of ROA and ROE of selected universities in Poland.
4) Thank you for your valuable comment and recommendation, which inspired us to conduct further, more in-depth research on the effect of the combined impact of all success factors for public universities in Poland but because of the very short time for improving the article, this method is not included.
Reviewer 3 Report
The article is well-founded and logically structured. A structural analysis was made. It gives unambiguous interpretations of univariate distributions. It is necessary to more clearly present information about the respondents, the period of the respondents and the survey method.
Author Response
Thank you for your valuable suggestion, based on which more detailed information about the respondents and the research method has been completed in the article (text marked in yellow). The research period was presented in the body of the article, i.e. from 8 May 10 June 2020.
Reviewer 4 Report
This work has a very interesting focus. It has a good structure and organisation. However, it does not seem to be of an innovative scientific nature, but just a detailed report on the topic addressed.
Major comment:
"The authors acknowledge the need for parametric and non-parametric methods to fully evaluate success."
This is a summary of the reasons for the non-acceptance of this paper as it is. The lack of a more in-depth statistical analysis is the decisive factor in order to give a strong scientific evidence.
Some minor remarks:
1) Figure 3. - Some items are referred with number but the results are percentages?
2) Line 609- "other indicators" - Please, give more details.
3)Line 255 - double comma
4) RQ1 answer - Line 522 - authors believe .... - It needs a detailed justification.
It needs a minor general proofreading.
Author Response
Thank you for your valuable feedback. The authors would like to point out that, for the purposes of this article, they have developed an author-consciously objective tool based on measurable data only.
1) Figure 3 shows the % distribution of responses of experts employed at public higher education institutions in Poland to the questions asked.
2) In response to the comment regarding line 609, the authors propose other indicators such as: ROI - Return on Investment and ROS – Return on Sales. Previously line 609, in the current version of the article.
3) We would like to thank you for your attention and clarify that Polish inverted commas were mistakenly used in the text of the article (line 255).
4) Thank you for your comment. We would like to clarify that the paragraph was mistakenly inserted here and has now been removed.
Round 2
Reviewer 1 Report
The content of the article has improved, although it remains unclear how the proposed results-oriented measurement tool can help leaders of public HEIs in strategic decision-making, and maintaining a good level of economic efficiency demonstrates the achievement of the policy objectives of sustainable development of public HEIs in Poland. It must influence the achievement of sustainable development objectives and the improvement of sustainability.
Author Response
Thank you for your valuable insight. The authors would like to point out that the article (lines 646 to 658) explains success levels and recommendations that can be used by public university leaders in their decision-making process.
Authors added detailed explanation - text in yellow (lines 662-669).
Authors added the explanation – yellow text (lines 441-443) and summing up (lines 702-723).
Reviewer 4 Report
Overall, this document has been improved, although the lack of strong statistical analysis is detrimental.
Some minor remarks:
Line 105 - Plu mmer
Line 320 - Actuarial
Line 341 to line 351 - Several categories were defined - A comparison has been done?
Figure 3 - The items in the chart should be uniformized (number or percentages). It is suggested delete number or percentage at the beginning of each item.
Figure 4 - Please, explain with U10 didnt follow the tendency.
Figures 5, 6 and 7 - ROA i ROE , you mean ROA and ROE
Line 695 - The last paragraph is very vague. It needs further development.
Please, a new proofreading is needed.
Author Response
The authors would like to clarify that the authors intended the expert opinions to be more of a starting point for further analysis and to provide a background to the research conducted. Therefore, there are no references to strong statistical analyses in the article.
The authors thank you for your observation (lines 105 and 320, Figures 4-7 ). The changes have been made in the text of the article.
The authors would like to clarify that, due to the limited number of characters, such a comparative analysis was not included in this article. Furthermore, the authors intended the experts' opinions to be rather a starting point for further analyses, and the statistical analysis used was the background for the research conducted.
The authors would like to clarify that the item names in the body of Figure 3 (numbers and percentages) are the proper names of the factors (KPIs) from the rankings that were used to identify the critical performance indicators (CPIs) by the respondents.
Authors added the explanation – yellow text (lines 441-443).
The change has been made in the text of the article (lines 702-723).